# Maternal Blood Levels of Toxic and Essential Elements and Birth Outcomes in Argentina: The EMASAR Study

**DOI:** 10.3390/ijerph19063643

**Published:** 2022-03-18

**Authors:** Shanshan Xu, Solrunn Hansen, Kam Sripada, Torbjørn Aarsland, Milena Horvat, Darja Mazej, Marisa Viviana Alvarez, Jon Øyvind Odland

**Affiliations:** 1Department of Public Health and Nursing, Norwegian University of Science and Technology (NTNU), 7491 Trondheim, Norway; shanshan.xu@uib.no; 2Center for International Health, Department of Global Public Health and Primary Care, University of Bergen, 5009 Bergen, Norway; 3Department of Health and Care Sciences, UiT The Arctic University of Norway, 9037 Tromso, Norway; solrunn.hansen@uit.no; 4Centre for Global Health Inequalities Research (CHAIN), Norwegian University of Science and Technology (NTNU), 7491 Trondheim, Norway; kam.sripada@ntnu.no; 5Centre for Digital Life Norway, Norwegian University of Science and Technology (NTNU), 7491 Trondheim, Norway; 6Research Department, Stavanger University Hospital, 4068 Stavanger, Norway; toaa@lyse.net; 7Department of Environmental Sciences, Jožef Stefan Institute, 1000 Ljubljana, Slovenia; milena.horvat@ijs.si (M.H.); darja.mazej@ijs.si (D.M.); 8Pediatric Department, Hospital Público Materno Infantil de Salta, Sarmiento 1301, Salta 4400, Argentina; marisavivianaalvarez@gmail.com; 9School of Health Systems and Public Health, Faculty of Health Sciences, University of Pretoria, Pretoria 0002, South Africa

**Keywords:** biomonitoring, children’s environmental health, environmental exposures, social determinants of health, toxic metals

## Abstract

Pregnant women’s levels of toxic and essential minerals have been linked to birth outcomes yet have not been adequately investigated in South America. In Argentina, *n* = 696 maternal whole blood samples from Ushuaia (*n* = 198) and Salta (*n* = 498) were collected in 2011–2012 among singleton women at 36 ± 12 h postpartum and analyzed for blood concentrations of arsenic (As), cadmium (Cd), mercury (Hg), lead (Pb), copper (Cu), manganese (Mn), selenium (Se) and zinc (Zn). This study examined the associations between maternal elements levels and birth outcomes, and sociodemographic factors contributing to elements levels. Maternal age, parity, body mass index, smoking, and education were linked to concentrations of some but not all elements. In adjusted models, one ln-unit increase in Pb levels was associated with increased gestational age (0.2 weeks, 95% CI = 0.01–0.48) and decreased birth weight (−88.90 g, 95% CI = −173.69 to −4.11) and birth length (−0.46 cm, 95% CI = −0.85 to −0.08) in the Salta sample. Toxic elements concentrations were not associated with birth outcomes in Ushuaia participants. Birth outcomes are multifactorial problems, and these findings provide a foundation for understanding how the body burden of toxic and essential elements, within the socioeconomic context, may influence birth outcomes.

## 1. Introduction

Human exposure to metal contaminants is a problem of serious concern due to the ubiquitous presence, high environmental persistence, and the adverse health effects of cadmium (Cd), mercury (Hg), lead (Pb), arsenic (As), and other toxic elements [1]. Consequences of exposure to these toxic elements include increased risks of cancer, disturbances of the cardiovascular and central nervous systems, cognitive impairment, endocrine disorders, and so forth [2,3]. Due to the immaturity of the developing immune system and the rapid pace of growth, pregnancy is a particularly sensitive window for exposure to toxic elements and organic pollutants, many of which persist in the environment for decades [4,5]. Bioaccumulation of toxic elements can contaminate the food chains, and therefore the diet has become the primary source of exposure for the general population [2].

Despite the placental barrier, Pb, Hg, and As can transfer from mother to fetus via placenta or through breastfeeding and accumulate in fetal tissues [4,6]. During the gestation period, the fetus can be exposed to toxic along with beneficial essential trace elements such as copper (Cu), manganese (Mn), selenium (Se), and zinc (Zn) [7]. Essential elements status (both deficiency and excess) has been closely linked with the health and well-being of pregnant women and the growth and development of their offspring [8]. However, epidemiological findings around toxic elements exposure and birth outcomes are inconsistent. Some studies report inverse associations even at lower levels of toxic element exposure [9,10] while others observed no significant associations [11,12]. Low birth weight and small size at birth have been shown to be associated with health and development problems during infancy and throughout childhood as well as with implications for later health in childhood, adolescence, and adulthood [13,14].

Toxic elements exposure is a major and understudied public health concern for children in South America [15]. A few studies regarding toxic elements exposure in pregnant women have been conducted in Argentina, mainly in northern Argentina and limited to several toxic elements [16,17,18]. Information with respect to essential elements levels in blood among pregnant women is scarce. It is also urgent to explore the situation in the southern coastal Argentina as climate change is influencing the exposure to contaminants and lifestyle in coastal areas [19]. A comparison between the high-altitude population in the north of Argentina (Salta) and coastal people living in the southernmost areas of the world (Ushuaia) is warranted. Therefore, the present study in Argentina could provide an important complement to traditional toxicological studies that have already demonstrated the toxicity of these elements.

To the best of our knowledge, the present study is the first to report maternal whole blood concentrations of toxic and essential elements related to pregnancy in Argentina. The aim was to examine newborn birth outcomes associated with maternal levels of toxic and essential elements, and sociodemographic factors contributing to maternal elements levels. Regional differences (coastal vs. inland) and interactions with sex were also assessed.

## 2. Materials and Methods

### 2.1. Study Population and Data Collection

The EMASAR (Estudio del Medio Ambiente y la Salud Reproductiva; English: Study on the Environment and Reproductive Health) study took place in Ushuaia in the south and Salta in the north of Argentina. Descriptions of the study areas have been provided in detail elsewhere [20]. In brief, Ushuaia, the southernmost city in the world, is the capital of the Tierra del Fuego in Antártida e Islas del Atlántico Sur Province. Ushuaia has a subpolar oceanic climate where fishing, sheep farming, natural gas and oil extraction, and ecotourism are the main economic activities. The city of Salta is the capital of the northwestern, highland Salta province. Agriculture and related industrial activities are the main economic activities. Salta is relatively underdeveloped with large socioeconomic inequalities and widespread poverty while the socioeconomic standards in Ushuaia are the most thriving in the country [20].

UiT The Arctic University of Norway and Stavanger University Hospital in Norway were responsible for the project in partnership with the private Clínica San Jorge in Ushuaia and the Hospital Público Materno Infantil in Salta.

The study period was from April 2011 to March 2012. More than 90% of invited women consented to participate for a total of 717 women recruited at delivery; 698 women offered whole blood samples and were thus included in this study. Through personal interviews, participants completed questionnaires covering personal characteristics, socio-economic factors, previous children and breastfeeding history, environmental and health and lifestyle conditions, and dietary intake. Obstetrical information of current and previous delivery was acquired through medical records. Non-fasting maternal blood samples, height, and weight were obtained at 36 ± 12 h following delivery (median 1 day, range 0–3 days). For details about the study profile material and methods, refer to Økland et al. [20]. After excluding two twin births from Ushuaia, the present analysis is based on *n* = 696 maternal whole blood samples collected from Ushuaia (*n* = 198) and Salta (*n* = 498) from singleton births at 36 ± 12 h after delivery.

The study was approved by the Ethics Committee of the Salta Medical Association (#2010/7317) and the Ministries of Health in both provinces. The Norwegian Regional Committee for Medical and Health Research Ethics (REC North) approved the study (#2011/706) and it was conducted in accordance with the Helsinki declaration. Written informed consent was obtained from all participants.

### 2.2. Chemical Analysis and Quality Control

Non-fasting venous blood sample was drawn from the maternal antecubital vein with standard equipment into a BD Vacutainer^®^ for trace elements (HemogardTM/Royal Blue, Ref# 368381; plastic, 6-mL, with 10.8 mg K_2_ EDTA; Becton Dickinson, Plymouth, U.K.). The whole blood was transferred to 4.5 mL cryovials and stored at minus 20 °C at the local hospital pending shipping to Norway. In Norway, the biological samples were deposited in the EMASAR Biobank at the UiT The Arctic University of Norway at minus 35 °C until analysis. For analysis, the frozen samples were subsequently transferred to the Department of Environmental Sciences, Jožef Stefan Institute (JSI), Ljubljana, Slovenia.

Chemical analyses of maternal whole blood were done for a suite of eight elements, including the toxic elements As, Cd, Hg, and Pb, and essential elements including Cu, Mn, Se, and Zn. With the exception of Hg, all analyses were performed using an octopole reaction system (ORS) inductively coupled plasma mass spectrometer (7500ce, Agilent Technologies, Santa Clara, CA, USA) equipped with an ASX-510 Autosampler (Cetac). Briefly, an aliquot of 0.3 mL of blood sample was diluted ten times with an alkaline solution containing Triton X-100 and ethylenediaminetetraacetic acid disodium salt dihydrate (EDTA) in a contamination-free tube [21]. An aliquot of an internal standard solution containing scandium (Sc), gallium (Ga), yttrium (Y), and gadolinium (Gd) was added. For calibration, the standard addition procedure was performed. Instrumental conditions were: Babington nebulizer, Scott-type spray chamber, reaction cell gas helium, and isotopes monitored ^45^Sc, ^55^Mn, ^63^Cu, ^66^Zn, ^69^Ga, ^75^As, ^78^Se, ^89^Y, ^111^Cd, ^114^Cd, ^157^Gd, ^206^Pb, ^207^Pb, and ^208^Pb. Tuning of the instrument was performed daily using a solution containing lithium (Li), magnesium (Mg), Y, cerium (Ce), thallium (Tl), and cobalt (Co). Quantification of all isotopes was performed using one central point of the spectral peaks and three repetitions.

Analytical precision was for Pb, Mn, Cu, Zn, and Se, 5%, for As, 10%, and for Cd, 15%. Limits of detection for Cd, Pb, As, Se, Cu, Zn, and Mn, calculated as three times the standard deviations of the blank sample, were 0.05 ng/mL, 1 ng/mL, 0.1 ng/mL, 5 ng/mL, 5 ng/mL, 30 ng/mL, and 0.5 ng/mL blood sample, respectively. The reference material Seronorm Trace Elements Whole Blood L-1 (SERO AS, Billingstad, Norway) was used for every 12 samples to check the accuracy of the results.

### 2.3. Determination of Total Hg in Whole Blood

The concentration of total mercury in maternal whole blood was determined by the Direct Mercury Analyzer DMA-80 (Milestone Srl, Sorisole, Italy). The system integrates thermal decomposition sample preparation, amalgamation, and atomic absorption detection. The following procedures were used: 100–200 mg of blood sample was weighed in a quartz boat and placed in an auto-sampler. Controlled heating in an oxygenated decomposition furnace was used to liberate mercury from the sample in the instrument. The sample was dried and then thermally and chemically decomposed within the decomposition furnace at 650 °C. An oxygen stream passing through the tube carries the remaining decomposition products through the amalgamator that selectively traps mercury vapor, which was subsequently desorbed for quantization. Flowing oxygen carried the mercury vapor through absorbance cells positioned in the light path of a single wavelength atomic absorption spectrophotometer. Absorbance was measured at 254 nm as a function of mercury concentration [22]. Analytical precision was 7% for blood Hg. Limit of detection was 0.2 ng/mL. The reference material Seronorm Trace Elements Whole Blood L-1 (SERO AS, Norway) was used to check the accuracy of the results for total mercury in the blood.

### 2.4. Statistical Analysis

Geometric means (GM) with 95% confidence intervals (CI), minimum, maximum, and percentiles were used for descriptive analyses of the concentration of maternal whole blood elements. Differences in maternal and neonatal characteristics between the study sites were tested for significance using Mann–Whitney U test and chi-square test. The relationships between elements were explored using Spearman’s rank correlation analysis, presented with Spearman’s rho (*ρ*). Data analyses were performed using R software (version 4.0.2; R Project for Statistical Computing, Foundation for Statistical Computing, Vienna, Austria).

Before inclusion in the analysis, the concentrations of the elements were natural logarithm transformed for normalization. Multiple linear regression models were used to assess the effects of the following independent variables on the concentrations of the elements: maternal age (years), pre-pregnancy body mass index (BMI, kg/m^2^), parity (primiparous vs. multiparous), smoking during the last year (yes vs. no), highest completed education level (primary/secondary vs. tertiary/university), study sites (Ushuaia vs. Salta), and residence (urban vs. semi-urban/rural). The equation followed ln (elements) = β_0_ + β_1_ (study sites) + β_2_ (age) + β_3_ (BMI) + β_4_ (parity) + β_5_ (smoking) + β_6_ (education) + β_7_ (residence) + ϵ. The obtained regression coefficients β were transformed into relative changes (%) for better representation of the estimated effects. An additive unit change (*c*) in each variable results in a relative change in the estimated geometric mean of levels of the elements, which are equal to 100 × (e^β×c^ − 1)%, and the corresponding confidence intervals were calculated as 100 × (e^β±z1−α/2×SE(β)^ − 1)%. Where β and the standard error (SE) from multiple linear regression analysis and *c* is set as the interquartile range of explanatory variables; thus, setting *c* = 1 for the binary variable (parity, smoking, education, study sites, and residence) [23].

The effects of natural logarithm transformed maternal elements levels on the birth outcomes (gestational age (weeks), birth weight (grams), birth length (cm), and head circumference (cm)) were assessed by multiple linear regression models. Results were presented as the change in birth outcomes per ln unit change in maternal elements levels of whole blood. We used directed acyclic graph to identify the potential confounders to include in the statistical model (Appendix A, Appendix A). The final models were controlled for maternal age, pre-pregnancy BMI, parity, smoking, and education. In addition, in order to tease out the direct effect of maternal blood elements levels on the birth outcomes, birth weight and birth length were added into the gestational age regression model, whereas gestational age was introduced into the birth weight, birth length, and head circumference models. Regional specific models were fitted for Ushuaia and Salta.

Logistic regression analysis was used to examine the associations between the tertile of maternal elements concentrations and binary outcomes including preterm birth (defined as gestational age < 37 weeks) and low birth weight (defined as birth weight < 2500 g). The criteria for selecting and retaining confounders and covariates in the logistic regression were similar to those for linear regression. Preterm birth was introduced into the low birth weight model and low birth weight was included in the preterm birth model [24]. Adjusted odds ratios and 95% confidence interval were used to report the relationships.

To assess the effect of the modification of the relationship between elements levels and birth outcome by infant sex, we refitted the multiple regression models with the addition of a two-way interaction term between sex and elements concentration, and the interaction term coefficient was tested for significance. A significance level of *p* < 0.05 (two tailed) was used for all analyses.

## 3. Results

### 3.1. Sociodemographic Characteristics

Maternal and child characteristics are summarized in Table 1. Demographic characteristics of 696 participating women in our study were described previously [20]. The mean pre-pregnancy BMI was 23.5 kg/m^2^ in both study sites, and the smoking rate prior to pregnancy and proportion of urban residents was similar between Ushuaia and Salta. Mothers in Ushuaia had a mean age 4 years older than those in Salta (*p* < 0.001) and had higher educational attainment. More than half of the women were multiparas. The average gestational age was close to 39 weeks at both sites, among which 32 (4.9%) were preterm birth. Infants from Salta weighed on average 90 g less (*p* = 0.016), had shorter birth length, and smaller head circumference than those from Ushuaia (*p* < 0.001). A total of 23 (3.3%) newborns had low birth weight.

### 3.2. Detection Frequency and Concentrations of Elements in Whole Blood

Descriptive statistics of the toxic and essential elements stratified by study sites are shown in Table 2. Both As and Pb, and all the essential elements, were detected in 100% of the study participants. Cd was found in almost all participants’ blood, and Hg in 97% of those from Salta and in 75% of those from Ushuaia. Significant regional differences were observed for all elements except Cd (*p* = 0.901). Blood geometric mean levels of Cu, Mn, Se, Hg, and Pb were significantly higher for Saltanean women compared with Ushuaia participants while the blood levels of Zn and As were the opposite. In both sites, those who reported smoking during the last year had higher levels of Cd (*p* < 0.001) compared with non-smokers. Appendix A shows the distribution of elements in whole blood of the overall sample.

### 3.3. Associations between Elements in Whole Blood and Maternal Characteristics

Appendix A and Figure 1 show the results of associations between maternal characteristics and whole blood elements levels. Significant differences between study sites persisted after the multivariate adjustment for other covariates. In addition, blood Cd concentrations were observed higher in Salta than in Ushuaia (*p* = 0.035, Appendix A). Whole blood levels of Cu, Cd, and Pb were elevated with increasing age but blood concentrations of Mn decreased with age (Figure 1). Parity was the main determinant for essential elements, showing positive relationships with blood levels of Mn, Se, and Zn and inverse association with Cu (Figure 1). No parity dependence was found in the toxic elements group across primiparous and multiparous. Higher BMI was associated with decreases in the levels of As (*p* = 0.032) and Cd (*p* < 0.001), and higher tobacco consumption prior to pregnancy was positively associated with concentration of Cd (*p* < 0.001) and Pb (*p* = 0.016) (Figure 1, Appendix A). Maternal education was significantly negatively associated with the levels of Cu and Mn. Living in an urban or semi-urban/rural location did not influence concentrations of elements.

### 3.4. Birth Outcomes and Maternal Whole Blood Elements Concentrations

Appendix A shows the results of adjusted associations between whole blood elements concentrations in the overall sample and gestational age or birth weight, length, and head circumference. Figure 2 presents the regional-specific adjusted associations. Whole blood Cu concentrations were associated with decreased birth weight, birth length, and head circumference in Saltanean women (Figure 2(B-1),C,D). Blood Se levels showed negative associations with birth length for both sites; and maternal blood Se concentrations were found to be associated with increased gestational age (β = 1.1, 95% CI = 0.14–2.0) in Ushuaia women and decreased birth weight in Salta participants (β = −422.8, 95% CI = −660.6 to −185.0) (Figure 2(A-1,B-1)). Blood Zn concentrations were inversely associated with birth weight, length, and head circumference in Saltanean participants (Figure 2(B-1),C,D).

For blood As and Cd levels, no significant associations with birth outcomes were found in this study. One ln-unit increase in blood Pb levels was found to be associated with increased gestational age (0.2 weeks, 95% CI = 0.01–0.48) and decreased birth weight (−88.90 g, 95% CI = −173.69 to −4.11) and birth length (−0.46 cm, 95% CI = −0.85 to −0.08) in the Salta sample (Figure 2(A-1,B-1),C). Maternal Hg concentrations did not present significant associations with birth outcomes for Ushuaia or Salta, except for the negative association with head circumference in Ushuaia women (−0.30 cm, 95% CI = −0.58 to −0.004) (Figure 2D). No clear dose–response relationships between tertile of blood elements levels and preterm birth or low birth weight were evident in the adjusted logistic regression analyses in our study (Figure 2(A-2,B-2)).

Appendix A, Appendix A, presents the specific estimated effects between elements concentration and birth outcomes in Ushuaia and Salta and Appendix A shows the details of the logistic regression analysis.

### 3.5. Sex Interaction

When the interactions between infant sex and elements levels were added into the model, the interaction term was significant for Cu and Zn. Stratified analysis by infant sex showed that the effects of Cu on gestational age (Figure 3(A-1)) and birth length (Figure 3B) were only significant among female infants (*p* < 0.001 for both models); similarly, the effect of Zn on gestational age (Figure 3(A-2)) was also only significant for female newborns (*p* = 0.024) (data not shown). The association between elements concentration and preterm birth and low birth weight stratified by infant sex were similar to the main logistic analyses results; no sex interactions were found.

### 3.6. Inter-Element Correlations

The regional-specific Spearman correlation between the eight different elements is presented in Appendix A. In both study sites, the strength of the significant positive correlations between the different whole blood elements concentrations was weak (*ρ* < 0.3), except a modest relationship between Zn and Se (*ρ* = 0.58 and *ρ* = 0.49 in Ushuaia and Salta, respectively). In Salta, Hg was significantly unrelated to all other elements, but in Ushuaia, it correlated to Se and As. In contrast, both As in Ushuaia and Se in Salta were related to nearly all elements, except for Cd and Hg, respectively. Pb in Salta was correlated with all elements except for Cu and Hg, which is in contrast to Pb in Ushuaia which only related to As, Cd, and Zn. Cd was also correlated with Zn; Se and Mn (Salta only); Cu also with Se; and Mn to Zn and Se (Salta only). Appendix A shows the inter-elements relationships of the whole sample.

### 3.7. Comparison with Other Studies

In general, the toxic element concentrations (As, Cd, Hg, and Pb) for the Ushuaia and Salta studied participants were in the middle to lower range. The concentrations of Mn and Zn in Ushuaia and Salta were in the middle to higher range. The blood Cu levels in Salta and blood Zn concentration in Ushuaia were higher than levels reported by other studies (Table 3).

## 4. Discussion

The present study is the first to our knowledge to report maternal whole blood concentrations of toxic and essential elements related to pregnancy in Argentina. Our results demonstrate that region of residence was a main determinant for blood element concentrations. The observed different maternal blood concentrations of toxic and essential elements between Ushuaia and Salta may reflect regional dissimilarities in, e.g., historical and current emissions through industry and agriculture, environmental soil conditions, and dietary intake. Further, factors such as parity, age, BMI, and level of education influenced the concentrations of several elements. Smoking was significantly positively associated with blood Cd and Pb levels. Certain elements showed significantly positive associations with gestational age and inverse relationships with birth weight, birth length, and head circumference. Logistic regression analysis revealed no evidence to suggest a clear dose–response relationship between maternal blood elements levels and preterm birth or low birth weight. Additionally, our findings suggest that female newborns may be more sensitive to maternal blood Cu and Zn levels. In general, essential elements such as Cu, Mn, Se, and Zn were considered sufficient and concentrations of the toxic metals As, Cd, Hg, and Pb were in the middle to lower range compared with other countries (Table 3).

### 4.1. Toxic Elements Concentrations and Factors Associated with Whole Blood Toxic Element Concentrations

#### 4.1.1. Lead (Pb)

Blood lead levels in Argentina have declined substantially following the national ban on leaded gasoline in 1996, measured both in children’s blood [43,44] and cord blood [17]. However, the United Nations Children’s Fund (UNICEF) recently estimated that lead exposure costs Argentina 0.91% of its gross domestic product per year due to lead’s profound and long-lasting consequences on child development [45]. Argentina Ministry of Health implemented 50 µg/L as the intervention level of blood Pb for children and pregnant women in the guideline in 2013 [46]. Eleven (1.6%) participating mothers had blood Pb levels higher than 50 µg/L in the present study. In general, the observed blood Pb concentration in Salta may be designated as moderate when compared with those reported for other regions (Table 3). By comparison, blood Pb concentrations of Ushuaia women were in the lower range and comparable with those reported in previous studies of pregnant women [26,34,37,40], although higher than those from Puerto Rico [33] and Canada [35]. The Pb concentration in the city Salta in our study revealed lower blood Pb levels than another study conducted in San Antonio de los Cobres, a town in Salta Province [18]. San Antonio de los Cobres has been recognized for elevated concentrations of lithium, boron, cesium, and Arsenic in drinking water [47]. It is likely that people in that area were exposed to more contaminants than other regions in Salta Province.

The positive association between age and Pb concentrations in this study was in agreement with previous studies on maternal measurements [35,48]. Older people may have been exposed to more Pb on average in their lifetime compared with younger people. Furthermore, as Pb is stored long term in bone, with a half-life ≥ 10 years [49], accumulative body burden of Pb increases over time. Smoking was associated with elevated blood Pb levels. Active and passive smoking both showed an increased body burden of Pb during pregnancy, which has been found previously [35] as tobacco and cigarette smoke contain Pb as the tobacco plants absorb Pb from the soil [50]. This was also reinforced by the correlation between Cd and Pb concentrations observed in this study. 

#### 4.1.2. Mercury (Hg)

Blood total mercury level ≥3.5 µg/L may be associated with increased risk to the developing fetal nervous system [51]. In our study, 1% of samples were ≥3.5 µg/L, all of whom lived in Salta. In general, Ushuaia women in our study had a very low blood Hg concentration compared with other reports from South America and other regions worldwide (Table 3). Blood Hg concentrations in Salta were similar to levels reported in Brazil [30], USA [34], Canada [35], and South Africa [38]. Mothers from Salta had a significantly higher level of Hg than those from Ushuaia. This was consistent with a study in Suriname, where they reported that mercury levels in hair were significantly higher in pregnant women from inland than the coast [52]. The difference may reflect industry activities and historical emissions in Salta that polluted ecosystems. The World Health Organization (WHO) has identified that dietary intake is the primary source of Hg exposure in the general population, where fish and fish products are the dominant sources of Hg exposure in the diet [53]. Women in Salta had significantly more frequent intake of freshwater fish than those in Ushuaia (23% vs. 17%) and negligible marine dietary intake compared with Ushuaia (<3% vs. 30%, respectively) [20]. Some Argentinian studies have reported that freshwater species contain Hg due to the polluted lakes [54,55], which could be an explanation for why the inland area had a higher level of blood Hg. However, the present study did not adjust for food consumption, and this might weaken the results somewhat. Study region was the only factor associated with concentration of whole blood Hg. This suggests that there were other sources of Hg that were not included in our study. In addition, an unrelated correlation of Hg to other elements in the Salta sample might suggest a specific local source different from the indicated common source of Hg, Se, and As in Ushuaia, namely, marine seafood [37]. In Northern-Norwegian pregnant women with relatively high marine dietary intake, Hansen et al. observed a modest relationship (*ρ* ~ 0.5) between Hg/As and Hg/Se and with each of the elements associated with regular marine fish and seafood intake [37], which was consistent with our correlation results from coastal Ushuaia participants. Occupational exposure to mercury, for example through hazardous recycling of e-waste [56,57], is another potential exposure route but was not explored in this study.

#### 4.1.3. Arsenic (As)

Exposure to As through groundwater has been a major public health problem in multiple areas in Argentina, including Salta, the Chaco region, Córdoba, and the Pampas [58,59,60]. The observed As concentration in Ushuaia and Salta were mostly in the low range and similar to those reported in previous studies [30,38,41] (Table 3). Environmental pollution through mining may explain the elevated levels in neighboring Bolivia [27]. The blood As concentrations in Salta women were about one fourth of those observed in San Antonio de los Cobres [18]; this might reflect elevated levels of As in drinking water in San Antonio de los Cobres resulting in increased blood As concentrations [58]. Furthermore, seafood intake dominated by the non-toxic organic form arsenobetaine typically enrich human As concentrations [61]. As it can be cleared from the blood within a couple of hours after it is absorbed [62], the blood concentrations reflect recent intake where the barely higher As in Ushuaia compared with Salta may be linked to higher marine fish intake [37]. The correlation between As and Se in Ushuaia was slightly stronger than those in Salta which was also likely to reflect the fish intake differences between inland and coastal areas. The increased maternal BMI was associated with declining blood As concentration in this study. However, there have been contradictory reports on the effects of BMI and blood As concentrations [63,64].

#### 4.1.4. Cadmium (Cd)

The concentrations of all women in the selected global comparison were far below the upper limit for Cd in blood of 5 µg/L recommended by the CDC (Table 3) [65]. The observed Cd concentrations were in a lower range for Ushuaia and Salta, and similar to the levels reported previously [18,34,35,37], but higher than those observed in Brazil and Colombia [26,30]. Our suggested increase in Cd blood concentration with age and smoking is in agreement with previous studies of Cd biomarkers among pregnant women [66,67,68]. Cd is mainly stored in the kidneys and liver in the body and is eliminated through urine with a limited daily excretion. Cigarette smoke contains Cd [69]. Although only a minority of the women reported smoking at delivery—4.5% and 9.6% in Ushuaia and Salta, respectively, in this study [20]—the revealed positive effect of smoking during the last year implies a long term effect of Cd exposure. Furthermore, we found a negative association between blood Cd levels and BMI and this is in line with other studies [70,71]. By contrast, several studies found null association of blood Cd levels and BMI [68,72]. The exact biological mechanism of BMI on Cd exposure requires further investigation.

### 4.2. Essential Elements Concentrations and Factors Associated with Essential Element Concentrations

In general, our observed whole blood levels of essential elements were in the middle to higher range compared with other studies (Table 3). These elements concentrations were considered sufficient for both sites. Except for the negative association between parity and Cu, parity was a positive determinant for Mn, Se, and Zn in this study. Previous studies on the association between parity and essential elements are rather controversial. Rodrigues et al. reported a positive relationship between parity and cord blood Mn levels during pregnancy [73]; inverse associations were found between parity and blood Zn and Se levels in Swedish women [74]; while Wasowicz et al. suggested that parity has no influence on the levels of Se, Zn, and Cu in Polish women [75]. Potential biological mechanisms underlying the effects of parity on essential elements concentrations are worthy of further investigation. Higher maternal education level is associated with lower blood Cu and Mn concentrations. Amorós et al. suggested that women belonging to the lowest social class had the highest Cu concentrations [76]. This was somewhat consistent with our finding that women with secondary education and below had a higher level of Cu than the women with tertiary and university education in this study. Education has been used as proxy for socioeconomic status because it affects both income and occupation [77]. Similarly, Rodrigues et al. reported that higher education and incomes were associated with decreased Mn levels [73]. However, no such associations were found between socioeconomic status, maternal age, and Cu level in Nepali pregnant women [78]. Aside from parity and education, higher blood Cu concentrations were increased with maternal age; this is in line with the finding of the INMA study in Spain [76], but Alverez et al. did not find such an association between Cu, Zn, and Se with maternal age [79]. We also found a negative association between maternal age and blood Mn levels but no such association was found in a Japanese population [67]. As the primary source of essential elements is diet, the most important determinant of blood essential elements levels may stem from the different patterns of food consumption of the participants.

### 4.3. Associations between Maternal Blood Elements Concentration and Birth Outcomes

#### 4.3.1. Toxic Elements

This EMASAR study provides evidence of associations between exposure to toxic and essential elements and birth outcomes. We found positive associations between Pb concentrations and gestational age in Salta women. Significantly higher blood Pb levels were reported in women in their third trimester than those in the first trimester of pregnancy [37,80]. Previous studies have reported a U-shaped curve in maternal blood Pb concentrations during pregnancy [48,81]. Many studies have shown an association between blood Pb levels and calcium needs, as Pb can mimic or compete with calcium in biological processes [82]. The peak demand for calcium is in the third trimester, and maternal response to meet this demand may allow calcium resorption from bone, especially when dietary calcium intake is inadequate [83]. Thus, pregnant women who were exposed to Pb prior to pregnancy may mobilize Pb from bone to blood, increasing maternal blood Pb levels in late pregnancy. In contrast, previous studies have found a negative association [66,84] or no association between Pb levels and gestational age [85,86]. Our results suggested that blood Pb exposure was associated with lower birth weight and shorter birth length in the Salta sample. Consistent with other studies, maternal blood Pb levels were inversely associated with neonatal anthropometry [10,81]. However, no significant associations between Pb concentration and anthropometric variables have been reported in other studies [12,85].

Maternal blood Hg concentration showed a significant positive association with gestational age and negative relationships with birth weight, length, and head circumference in the entire sample. These findings were in agreement with previous studies [87,88,89], while no association was reported in other epidemiological studies [11,66,90]. Interestingly, despite higher blood Hg concentration in Salta women, these associations became insignificant after stratification by region, with the exception of the association between Hg levels and head circumference in the Ushuaia sample, indicating that infants in Ushuaia may be more affected by Hg than those in Salta, even at a lower level. We did not find an association between either As or Cd blood concentrations and gestational age and neonatal anthropometry, which was inconsistent with the previously reported inverse association between As and Cd exposure and birth outcomes [90,91]. These conflicting results across studies suggests that further research is needed to clarify the relationship between toxic elements exposure and intrauterine growth.

#### 4.3.2. Essential Trace Elements

Cu was an important exposure associated with birth outcomes in the present study. We found an association between higher levels of Cu in maternal blood and longer gestation length in the overall sample and lower birth weight, shorter birth length, and head circumference in Salta women. Due to the development of the placenta during pregnancy, blood Cu levels increasing in the late gestational stages [92]. The safe level of Cu for adequate fetal development was unknown. Cu deficiency and excess could contribute to adverse effects for the fetus and child development [93]. Studies reporting a possible association between Cu levels and neonatal anthropometric variables are relatively limited. Ozdemir et al. suggested birth weight was negatively correlated with maternal Cu level, which was in agreement with our results [94]. Several studies reported that blood Cu levels were associated with preterm birth [93,95]. Further studies about Cu effects on birth outcomes are warranted. We found that associations between Cu and gestational age and birth length varied by sex, such that blood Cu was positively associated with gestational age and negatively associated with birth length among female infants. Ashrap et al. suggested that no such sex-specific association was found between blood Cu concentrations during the second trimester and birth outcomes in Puerto Rico [66]. The different sampling time points may explain the differences in the sex effects on the birth outcomes. However, the underlying mechanism of these varied sex differences needs to be further elucidated.

Apart from Cu, Se is the main essential trace element associated with birth outcomes that showed negative relationships with neonatal anthropometric measurements. Our negative findings with increased Se were contrary to several studies that reported positive associations between cord and maternal blood and placenta Se levels and birth weight and birth height [68,96,97]; and no association between blood Se and birth size was reported by Kobayashi et al. [89]. Regarding the relationship between Se status and birth outcomes, there is still much disagreement between present results and existing literature, and further study is required to identify these relationships.

A few studies have investigated the links between maternal Zn status and birth outcomes. We observed negative relationships between Zn levels and birth weight, birth length, and head circumference in the Salta sample. Similarly, McMichael et al. suggested that maternal mid-pregnancy zinc status was weakly negatively correlated with length of gestation and birth weight [98]. However, most of the previous studies have reported a positive association between maternal blood Zn levels and birth weight [99,100], and some studies suggested that no significant association was found between Zn concentration and fetal outcomes [101,102]. Recently, Ashrap et al. found that blood Zn was negatively associated with gestational age among female newborns, but not males [66]. In contrast, our study suggested maternal Zn levels were positively associated with the gestational age among female infants. There are progressive physiological changes in the body throughout pregnancy. Varied blood Zn levels and association with birth outcomes may reflect the different pregnancy processes, including oxidative stress, inflammation, and other key functions [66,103] which can play a critical role in the gestational length. These discrepant results suggest further study on the mechanisms of this sex difference is warranted.

No significant associations between Mn exposure and birth outcomes were observed in this study. Similarly, several previous studies demonstrated that maternal Mn concentrations are not associated with birth outcomes [104,105]. Zota et al. suggested that birth weight increased with Mn levels up to 31 µg/L, but lower birth weight at levels above 31 µg/L [106], while Chen et al. reported Mn levels up to 41.8 μg/L with a decrease in birth weight [107]. Further investigation is necessary to determine whether an Mn exposure threshold exists.

### 4.4. Study Strengths and Limitations

The findings in the present study address the knowledge gap of maternal toxic and essential trace elements exposure in Argentinian pregnant women in two geographical areas. Demographic, socioeconomic, and lifestyle characteristics that differ between coastal inhabitants and high-altitude populations in Argentina were captured by this study and, together with biomonitoring and birth outcomes data, can offer novel insights in this area of maternal environmental health.

Despite the strengths, the present study also has some limitations. The sample size is relatively small with some results showing large confidence intervals, and it is therefore important that our study results are replicated in larger studies among different sample groups. There were several residual confounding factors that were not considered, such as frequency of food intake, vitamin and folic acid supplements, and other interactions. Self-reported data including pre-pregnancy weight, smoking habits, and education might have contributed to bias. Maternal blood samples were only measured once after the delivery, which may not accurately reflect elements concentrations during the pregnancy and postpartum. Previous studies have reported a U-shaped curve in maternal blood lead concentrations during the pregnancy, for example, Nishioka et al. found that blood lead levels were decreased at 25 weeks and then increased at 36 weeks of gestation [81]. A study from Northern Norway found that blood Zn, Mn, Pb, As, and Cd (non-smoker) concentrations increased during the pregnancy and from birth to 6 weeks postpartum [37]. Therefore, research is needed to analyze associations between blood elements levels in different trimesters as well as postpartum. Umbilical cord blood samples were not collected in the EMASAR study. Although significant associations between maternal blood element levels and umbilical cord blood elements concentrations have been demonstrated previously [27,36], measuring umbilical cord blood could more accurately indicate the redistribution of elements between mother and fetus. Observed associations between elements and birth outcomes might be confounded by mixtures with other toxic substances which were not included in this study. Further studies should consider the effects of mixtures exposure on birth outcomes.

## 5. Conclusions

The results of the present biomonitoring study provide novel insights relevant for public health in both northern and southern Argentina. The region of residence was one of the main determinants for both toxic and essential elements exposure. Maternal characteristics such as age, pre-pregnancy BMI, smoking, and education were also shown to predict maternal levels of certain elements postnatally. Delivering mothers in Salta had significantly lower education, showed higher levels of Cd, Hg, and Pb after adjustment for covariates, and nearly all toxic elements were detectable in Salta women, putting them and their children at risk. The differences between Ushuaia and Salta can be inferred to reflect differences in socioeconomic conditions, prior or current industry emissions, soil contamination, dietary intake, and other potential contributors of current and historical regional pollution. Birth outcomes are multifactorial problems, and these findings provide a foundation for understanding how the body burden of toxic and essential elements, within the socioeconomic context, may influence birth outcomes.

## Figures and Tables

**Figure 1 ijerph-19-03643-f001:**
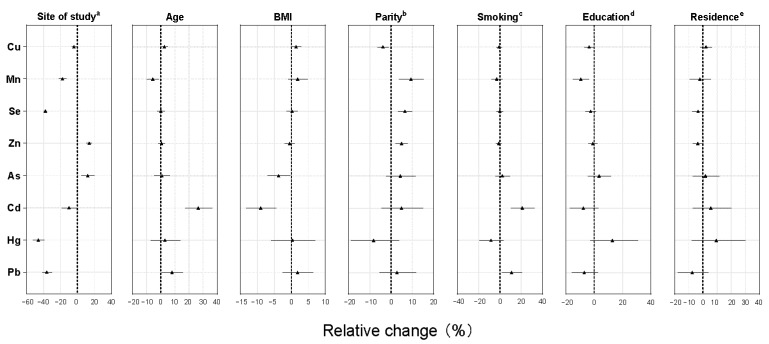
Relative change (%) in geometric mean of maternal whole blood concentrations (µg/L) of essential and toxic elements by unit change calculated from coefficients β and standard errors of the multiple linear regression analyses described in Appendix A, Appendix A. The unit of change for each variable was set as interquartile range. ^a^ Salta as reference site (Salta vs. Ushuaia); ^b^ para 1 as reference category (para 1 vs. parous); ^c^ non-smokers as reference (non-smoker vs. smoker); ^d^ the highest education level is primary or secondary as the reference group (primary/secondary education vs. tertiary/university); and ^e^ urban dweller as reference category (urban vs. semi-urban and rural). Abbreviations: As, arsenic; BMI, body mass index; Cd, cadmium; Cu, copper; Hg, mercury; Mn, manganese; Pb, lead; Se, selenium; and Zn, zinc.

**Figure 2 ijerph-19-03643-f002:**
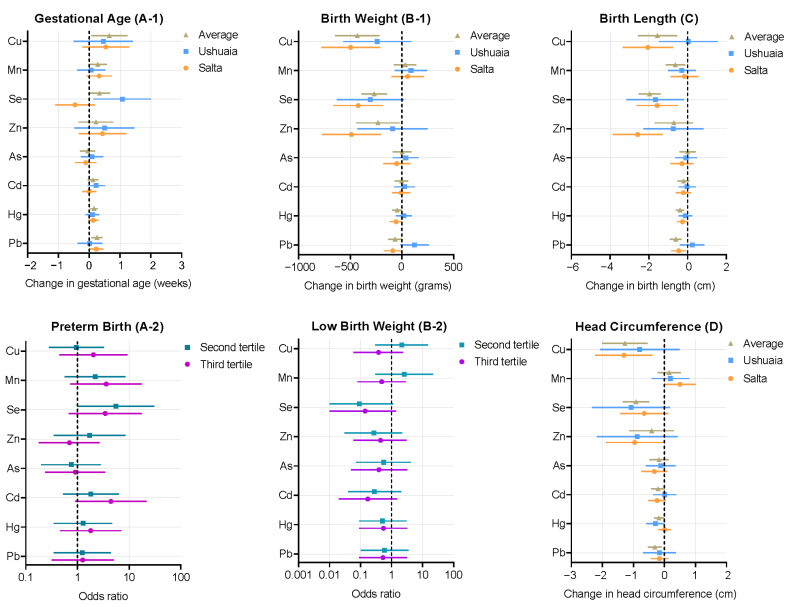
Changes in the birth outcomes associated with overall and regional specific blood elements concentrations (**A-1**,**B-1**,**C**,**D**). The results of logistic regression analyses show the relationships between tertile of maternal elements levels and preterm birth and low birth weight (**A-2**,**B-2**). The elements concentrations were natural logarithms transformed. Effect estimates presented as changes in the gestational age, birth weight, length, and head circumference for one ln-unit change in the elements. The odds ratios of the logistic regression analyses are displayed with 95% confidence interval, with the first tertile of the elements levels as the reference. All models were adjusted for maternal age, parity, pre-pregnancy BMI, smoking, and education. In addition, birth weight and birth length were added into the gestational age regression model (**A-1**), and gestational age was introduced into the birth weight (**B-1**), length (**C**), and head circumference (**D**) regression models. Low birth weight was introduced into preterm birth (**A-2**) and preterm birth was introduced into low birth weight model (**B-2**). Abbreviations: As, arsenic; Cd, cadmium; Cu, copper; Hg, mercury; Mn, manganese; Pb, lead; Se, selenium; and Zn, zinc.

**Figure 3 ijerph-19-03643-f003:**
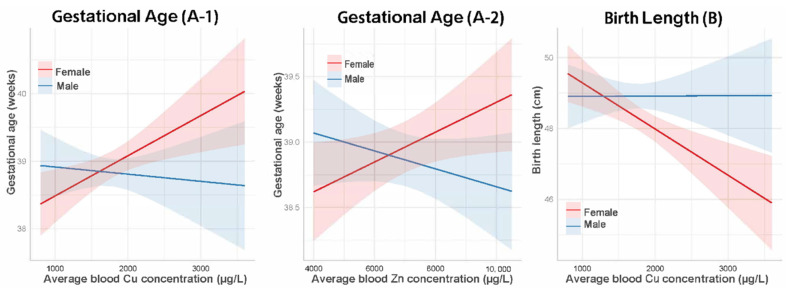
Interaction effects of infant sex on the association between average maternal Cu concentration and gestational age and birth length; and the relationship between average maternal Zn level and gestational age. Models (**A-1**,**A-2**) were adjusted for maternal age, parity, pre-pregnancy BMI, smoking, education, birth weight, and birth length. Model (**B**) was adjusted for maternal age, parity, pre-pregnancy BMI, smoking, education, and gestational age. Abbreviations: Cu, copper; Zn, zinc.

**Table 1 ijerph-19-03643-t001:** Characteristics of studied population in the EMASAR study 2011–2012.

Characteristic	N (Missing Data)	Ushuaia	Salta	*p* Value
All participants	696	198	498	
Maternal age (years), mean (SD)	696 (0)	28.8 (6.6)	24.7 (6.2)	<0.00 ^a^
Pre-pregnancy BMI (kg/m^2^)	637 (59)	23.5 (4.1)	23.5 (4.2)	0.884 ^a^
Parity	696 (0)			0.477 ^b^
Para 1, *n* (%)		82 (41.4)	221 (44.4)	
Parous, *n* (%)		116 (58.6)	277 (55.6)	
Smoking, last year	696 (0)			0.375 ^b^
No, *n* (%)		141 (71.2)	371 (74.5)	
Yes, *n* (%)		57 (28.8)	127 (25.5)	
Highest completed education level	695 (1)			<0.001 ^b^
Primary and secondary, *n* (%)		102 (51.5)	445 (89.5)	
Tertiary and university, *n* (%)		96 (48.5)	52 (10.5)	
Residence	696 (0)			0.087 ^b^
Urban, *n* (%)		181 (91.4)	432 (86.7)	
Semi-urban and rural, *n* (%)		17 (8.6)	66 (13.3)	
Gestational age (weeks), mean (SD)	658 (38)	38.8 (1.3)	38.8 (1.3)	0.579 ^a^
Birth weight (g), mean (SD)	687 (9)	3383.2 (438.9)	3292.9 (476.1)	0.016 ^a^
Birth length (cm), mean (SD)	688 (8)	49.7 (2.1)	48.5 (2.2)	<0.001 ^a^
Head circumference (cm), mean (SD)	686 (10)	34.9 (1.5)	34.3 (1.4)	<0.001 ^a^
Low birth weight, *n* (%)	23	5 (2.5)	18 (3.6)	0.463 ^b^
Preterm birth, *n* (%)	32	15 (7.6)	17 (3.4)	0.032 ^b^

^a^ Mann–Whitney U test. ^b^ Chi-square test.

**Table 2 ijerph-19-03643-t002:** Limits of detection and whole blood concentration (µg/L) of essential and toxic elements in Argentinian postpartum women living in Ushuaia and Salta (*n* = 696).

Elements	Ushuaia (*n* = 198)						Salta (*n* = 498)						*p* Value *
	% > LOD	GM (95% CI)	Min–Max	Selected Percentiles			% > LOD	GM (95% CI)	Min–Max	Selected Percentiles			
				25th	50th	75th				25th	50th	75th	
Cu	100	1688 (1646–1730)	961–2539	1515	1718	1885	100	1782 (1759–1804)	1057–3559	1616	1766	1959	0.001
Mn	100	17.83 (16.97–18.73)	4.75–39.53	13.74	18.11	23.06	100	22.98 (22.44–23.54)	8.65–52.96	19.28	22.60	27.61	<0.001
Se	100	80.06 (78.26–81.91)	51.28–125.06	71.19	80.56	89.00	100	129.25 (127.23–131.30)	75.36–240.57	114.27	128.83	145.35	<0.001
Zn	100	7633 (7468–7801)	4973–10,199	6767	7815	8665	100	6682 (6597–6768)	4171–9549	6049	6720	7343	<0.001
As	100	0.63 (0.59–0.67)	0.22–3.78	0.47	0.58	0.83	100	0.55 (0.53–0.57)	0.28–3.78	0.44	0.53	0.64	<0.001
Cd (Smoker)	98	0.22 (0.19–0.26) ^a^	0.04–0.90	0.17	0.20	0.33	99	0.21 (0.19–0.23) ^b^	0.04–1.05	0.15	0.21	0.28	0.501
Cd (Non-smoker)	95	0.17 (0.15–0.19) ^c^	0.04–0.51	0.12	0.18	0.26	99	0.18 (0.17–0.19) ^d^	0.04–0.76	0.13	0.18	0.25	0.394
Hg	75	0.35 (0.32–0.38)	0.1–2.7	0.1	0.3	0.6	97	0.62 (0.58–0.65)	0.1–19.6	0.4	0.6	0.9	<0.001
Pb	100	10.1 (9.5–10.7)	2.9–39.1	7.6	9.8	13.0	100	15.8 (15.2 −16.5)	3.9–152.3	11.8	15.0	20.9	<0.001

* Mann–Whitney U test. ^a,b,c,d^ The number of participants in each group is 57, 127, 141, and 371, respectively. Abbreviations: As, arsenic; Cd, cadmium; CI, confidence interval; Cu, copper; GM, geometric mean; Hg, mercury; LOD, limit of detection; Mn, manganese; Pb, lead; Se, selenium; and Zn, zinc.

**Table 3 ijerph-19-03643-t003:** Global Comparisons of blood elements concentrations among pregnant women or delivering women (µg/L).

	Location and References	Sampling Years	N	As	Cd	Hg	Pb	Cu	Mn	Se	Zn
Latin America and Caribbean	Ushuaia, Argentina ^a^ Present study	2011–2012	198	0.63	0.22 (smoker) (0.33) 0.17 (non-smoker) (0.18)	0.35 (0.34)	10.08 (9.81)	1688 (1781)	17.83 (18.11)	80.06 (80.56)	7633 (7815)
	Salta city, Argentina ^a^ Present study	2011–2012	498	0.55	0.21 (smoker) (0.21) 0.18 (non-smoker) (0.18)	0.62 (0.60)	15.83 (14.96)	1782 (1766)	22.98 (22.60)	129.25 (128.83)	6682 (6720)
	San Antonio de los Cobres, Argentina ^b^ [18]	2012–2013	169	2.2	0.16	-	21	-	-	86	6100
	French Guiana ^a^ [25]	2013	531	-	-	-	32.6	-	-	-	-
	Colombia ^b^ [26]	2009–2010	381	-	0.01	-	9.5	-	-	-	-
	Bolivia ^a^ [27]	2007–2008	419	6.14	-	-	26.53	-	-	-	-
	Peru ^a^ [28]	2004–2005	204	-	-	-	-	-	-	113.7	-
	Suriname ^b^ [29]	2016	76	-	-	3.88	47.3	-	-	-	-
	Brazil ^b^ [30]	2007–2008	155	0.6	0.09	0.6	16.2	1735	16.7	64	6420
	Mexico ^a^ [31]	2007–2008	299	-	-	-	23.8	-	-	-	-
	Costa Rica ^a^ [32]	2010–2011	418	-	-	-	-	-	23.5	-	-
	Puerto Rico ^a^ [33]	2011–2017	1183	0.34	0.12	1.2	3.3	1552	11.3	-	4682
North America	USA ^a^ [34]	2009–2011	211	0.45	0.18	0.45	8.9	-		-	-
	Canada ^b^ [35]	2008–2011	1673		0.20	0.56	5.60	-	12.64	-	-
Europe	Spain ^b^ [36]	2016–2017	40	1.8	0.4	1.8	12	1664	16	107	6708
	Norway ^a^ [37]	2007–2009	211	1.8	0.23 (smoker) 0.17 (non-smoker)	1.0	9.2	1780	15.8	72	5480
Africa	South Africa ^b^ [38]	2005–2006	62	0.57	0.15	0.65	23	1730	16.8	104	6290
	Benin ^a^ [39]	2015	60		0.35	-	38.0	1544	16.1		5215
Asia–Pacific	Japan ^b^ [40]	2001–2006	649	4.06	1.18	-	10.83	1289.2		176.4	4769
	China ^b^ [41]	2010	215	0.52	0.47	0.26	24.48	-	-	-	-
	Australia ^b^ [42]	2008–2011	173	1.26	-	-	-	1252	6.45	88.2	2330

The brackets present the median concentration of elements in the present study. ^a^ Values expressed as geometric mean in the study. ^b^ Values expressed as median in the study. Abbreviations: As, arsenic; Cd, cadmium; Cu, copper; Hg, mercury; Mn, manganese; Pb, lead; Se, selenium; and Zn, zinc.

## Data Availability

Not applicable.

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
