# Peer review of "Maternal Blood Levels of Toxic and Essential Elements and Birth Outcomes in Argentina: The EMASAR Study"

_ijerph, 2022, doi:10.3390/ijerph19063643_

Round 1

Reviewer 1 Report

This is an original article that analyzed association between eight elements and birth outcomes in Argentina. Overall, the authors demonstrated the associations between Pb and birth outcomes in Salta sample, but the results were different depending on each region. This article worth analyzing correlations between elements and birth outcomes by Argentina cohort.

The methods and statistics used are well described and the conclusions derived from these and the interpretations of the analysis are consistent, but I have a concern about covariates.

Since gestational age, birth weight, and birth length mean almost the same thing as the outcome (e.g., preterm infants are usually low birth weight), it is strange to add them in covariates. Gestational age, birth weight, and birth length should strongly correlate with each other, so the problem of multicollinearity also arises. Therefore, I suggest authors exclude them from covariates in the adjusted model. After excluding these from covariates, the results of logistic regression analyses can be different, so at least the excluded regression models should be added.

Further, when considering covariate selection or multicollinearity problems, directed acyclic graphs (DAGs) are useful.

Reviewer 2 Report

Thank you for the opportunity to read this relevant study. The authors studied maternal blood levels of toxic and essential elements during pregnancy and birth outcomes in Argentina. This is part of the EMASAR study that enrolled pregnant women during the period of 2011-2012 in two sites in Argentina (198 samples from Ushuaia and 498 samples from Salta).

 I have some comments.

Maternal exposure to toxic elements is a public health concern, but this study had data collection carried out more than 10 years ago. Could this long elapsed time interfere with the results presented? I have concerns about its applicability today, as habits or the environment may have changed. I believe that currently these cities had improved economic activities, mainly related to ecoturism that could influence the exposure today. Do the authors know whether these changes may interfere with the exposure of the local women?

I observed that the samples were collected at a private clinic in Ushuaia and at a public hospital in Salta. Do the authors believe that this difference in the profile may have influenced the results? Note that, in Ushuaia there was a significantly higher proportion of tertiary and university education and maternal education was significantly negatively associated with Cu and Mn levels. Could the education level have influenced nutrition and dietary mineral intake? Could women's habits be different and favor greater exposure to some metals?

The authors state that region of residence was a main determinant for blood element concentrations. Is this related to region of residence or educational level?

I believe that these doubts can be clarified by the authors.
